# Sportomics Analyses of the Exercise-Induced Impact on Amino Acid Metabolism and Acute-Phase Protein Kinetics in Female Olympic Athletes

**DOI:** 10.3390/nu16203538

**Published:** 2024-10-18

**Authors:** Renan Muniz-Santos, Adriana Bassini, Jefferson Falcão, Eduardo Prado, LeRoy Martin, Vinod Chandran, Igor Jurisica, L. C. Cameron

**Affiliations:** 1Lorraine Protein Biochemistry Group, Graduate Program in Neurology, Gaffrée e Guinle University Hospital, Rio de Janeiro 20270-004, Brazil; renanmuniz@edu.unirio.br (R.M.-S.); esprado@iefe.ufal.br (E.P.); 2Laboratory of Protein Biochemistry, The Federal University of the State of Rio de Janeiro, Rio de Janeiro 22290-250, Brazil; bassiniadriana@gmail.com; 3Laboratory for Research in Physical Exercise and Metabolism, Federal University of Alagoas, Maceió 57072-970, Brazil; jefferson.silva@cedu.ufal.br; 4Graduate Program in Nutrition, Faculty of Nutrition, Federal University of Alagoas, Maceió 57072-970, Brazil; 5Waters Technologies, Milford, MA 01757, USA; roy_martin@waters.com; 6Arthritis Program, Schroeder Arthritis Institute, Krembil Research Institute, University Health Network, Toronto, ON M5T 0S8, Canada; vinod.chandran@uhn.ca; 7Division of Rheumatology, Department of Medicine, Institute of Medical Science, Faculty of Medicine, University of Toronto, Toronto, ON M5S 1A8, Canada; 8Osteoarthritis Research Program, Division of Orthopedic Surgery, Schroeder Arthritis Institute and Data Science Discovery Centre for Chronic Diseases, Krembil Research Institute, University Health Network, Toronto, ON M5T 0S8, Canada; juris@ai.utoronto.ca; 9Departments of Medical Biophysics and Computer Science, Faculty of Dentistry, University of Toronto, Toronto, ON M5G 2L3, Canada; 10Institute of Neuroimmunology, Slovak Academy of Sciences, 845 10 Bratislava, Slovakia

**Keywords:** amino acid metabolism, elite female athlete, exercise immunology, acute-phase protein, Olympic Games, dried blood spot, mass spectrometry, sportomics

## Abstract

Background: Exercise can be used as a model to understand immunometabolism. Biological data on elite athletes are limited, especially for female athletes, including relevant data on acute-phase proteins and amino acid metabolism. Methods: We analyzed acute-phase proteins and amino acids collected at South American, Pan-American, and Olympic Games for 16 Olympic sports. We compared female and male elite athletes (447 vs. 990 samples) across four states (fasting, pre-exercise, post-exercise, and resting) to understand sex-specific immunometabolic responses in elite athletes. Results: Considering all states and sports, we found that elite female athletes exhibited higher concentrations of C-reactive protein, lipopolysaccharide-binding protein, myeloperoxidase, haptoglobin, and IGF1, with ratios ranging from 1.2 to 2.0 (*p* < 0.001). Women exhibited lower concentrations of most amino acids, except for glutamate and alanine. Although almost 30% lower in women, branched-chain amino acids (BCAAs) showed a similar pattern in all states (ρ ≥ 0.9; *p* < 0.001), while aromatic amino acids (AAAs) showed higher consumption during exercise in women. Conclusion: We established sex dimorphism in elite athletes’ metabolic and inflammatory responses during training and competition. Our data suggest that female athletes present a lower amino acid response towards central fatigue development than male athletes. Understanding these differences can lead to insights into sex-related immuno-metabolic responses in sports or other inflammatory conditions.

## 1. Introduction

Exercise-induced impacts on amino acid metabolism and acute-phase protein kinetics have been a topic of many investigations [1,2,3].

Amino acids (AAs) play multiple roles related to exercise, such as serving as energy sources, signaling protein synthesis during recovery, or facilitating neurotransmitter synthesis. AA supplementation in athletes has been extensively investigated as a performance-enhancing and recovery strategy, especially for glutamine, alanine, and the branched-chain amino acids (BCAAs; valine, isoleucine, and leucine) [4,5,6,7]. The role of amino acids in different central nervous system physiopathological conditions has been widely discussed [7,8,9,10]. A pivotal role of amino acid metabolism in developing exercise-induced hyperammonemia and central fatigue during exercise has also been investigated [8,11,12]. This effect seems to be caused by amino acid metabolism and AMP deamination through the myokinase reaction (E.C. 2.7.4.3) [13]. Both processes can result in the release of ammonia into the blood, exceeding its clearance capacity and subsequently impairing physical and mental functions due to ammonia’s toxic properties [14,15,16]. The altered proportion of amino acids in the bloodstream can change their concentrations within the central nervous system, potentially impacting neurotransmitter synthesis and leading to fatigue (as proposed by Fischer and Newsholme) [17,18]. AA profile changes can also reveal underlying regulation of biochemical pathways, such as gluconeogenesis, ketogenesis, and lipolysis [19,20]. Acute-phase proteins (APPs) may be upregulated (positive APPs) or downregulated (negative APPs) in response to inflammation-triggering events [21]. They have been considered markers indicating the systemic effects of cytokine regulation, which, conversely, are difficult to measure clinically due to their lower concentrations (~pg/mL) and short half-lives. Intense exercise can induce a state often called hypermetabolic stress and affect the inflammatory acute-phase response [22]. Assessing APPs’ response in exercise can shed light on hemolysis, gut permeability, bacterial translocation, and the innate immune response [23,24,25]. Select proteins not categorized as acute-phase proteins can be evaluated with APP analysis, providing other information regarding kidney function and volemia changes [23]. Understanding APPs’ kinetics in response to exercise can be an excellent model for understanding the immunometabolic responses in human and animal physiology and pathophysiology [26,27].

Dried blood spots (DBSs) can be used to carry out accurate and effective blood AA and APP measurements [23,28,29]. The convenient sample storage and collection from fingertip capillary blood in DBS have enabled multiple collections in real conditions faced by athletes [30]. DBS can be easily used to monitor athletes’ responses to different exercise and recovery phases and analyze the impact of different exercise sessions during training and competition, supporting individually tailored precision interventions.

It has been acknowledged that women may exhibit distinct metabolic responses to exercise compared to men [31]. There is evidence of sex-specific differences in the regulation of BCAA catabolism in mouse models [32], and various studies have revealed sex-based differences and suggested different interventions for male and female athletes [33]. Considering that energy metabolism is linked to amino acid metabolism, it is reasonable to hypothesize that amino acid and acute-phase protein metabolism could also show sex-specific differences in elite athletes. Indeed, professional athletes have been shown to exhibit different concentrations of metabolites, such as amino acids, compared to control individuals [34]. However, most studies on AA, APP, and other metabolic parameters have focused on male and non-elite athletes and neglected to investigate their responses to different sports in real competitions or training sessions [35,36]. In fact, data related to elite athletes and teams—not only metabolic data—are often not published due to privacy protection, competitive secrets, and lack of incentives or interest from clubs and federations [37,38].

More information concerning AA and APP metabolism in high-level sports needs to be provided, particularly for elite female athletes since increasing numbers of female athletes participate in Olympic events [39,40]. Data from the International Olympic Committee (IOC) show that the proportion of women participants in the Olympic Games has seen a notable rise—from 34% of the total in Atlanta in 1996 to a new high of 48% in Tokyo 2020, with a pledge to achieve complete sex equality in the Games of the XXXIII Olympiad in Paris 2024 [41]. It has been shown that female athletes are exposed to a higher risk of injury [39], which is also partially dependent on hormonal and metabolic changes related to the menstrual cycle [40]. In this sense, we previously detected gaps in the existing literature on the interplay between exercise metabolism and the menstrual cycle in elite female athletes [42]. Due to the lack of female-specific data, data retrieved from male athletes are used to manage women’s training and recovery, which is, at minimum, inefficient and potentially harmful.

Our main objective was to investigate AA and APPs’ responses in Olympic athletes from 16 Olympic sports across different states (fasting, pre-exercise, post-exercise, and resting) during training sessions and competitions. By performing sex-based stratification, we aimed to identify and characterize potential sex-specific differences and present all quantitative data. Due to the similarity of metabolic and inflammatory pathways in exercise and certain pathological conditions, our results may provide a better understanding of these responses in different scenarios.

## 2. Materials and Methods

### 2.1. Participants

Using our biobank of dried blood spot (DBS) samples collected during major sports events and training sessions of athletes from the Brazilian Olympic Committee (BOC), we identified all female athletes who had provided samples for amino acid (AA) and acute-phase protein (APP) analysis between January 2014 and January 2016. All procedures involving human subjects were approved by the ethics committee for human research at the Federal University of the State of Rio de Janeiro (117/2007, updated and renewed in 2011, 2013, and 2016) and the Federal University of Mato Grosso (2017–2021) and met the requirements regulating research on human subjects [43]). Participants were obligated not to use any prohibited performance-enhancing substances or methods, as confirmed by the negative results from various doping control analyses. Moreover, according to BOC policy, all athletes were advised not to use supplements, even from compounding pharmacies, due to the risk of contamination. We retrieved 54 female athletes and 272 samples for APP analysis, and 17 female athletes and 175 samples for AA analysis. The female athletes represented 11 different sports disciplines (athletics, boxing, cycling, modern pentathlon, karate, taekwondo, softball, gymnastics, basketball, archery, and swimming).

Using the same biobank, we identified all male athletes who had provided samples for AA and APP analysis during the same period. In total, we retrieved 76 male athletes with 471 samples for APP analysis, and 47 male athletes with 520 samples for AA analysis. The male athletes represented 16 different sports disciplines (athletics, boxing, beach volley, canoeing, cycling, handball, modern pentathlon, karate, taekwondo, softball, gymnastics, basketball, archery, diving, ultra-marathon, and swimming).

The athletes provided the blood mainly in four different states: fasting, pre-exercise or post-exercise (meaning before or after training or competition), and resting (mostly one hour after the end of exercise) for different sports, both women and men. The sample collections were performed under field-of-play conditions that the athletes face in their training sessions or competitions, i.e., an environment that cannot be controlled as in laboratory conditions. The competitions in which our collections were performed encompassed the Pan American and South American Games. Most of the athletes analyzed are also Olympic medalists.

### 2.2. Sample Collection and Amino Acid and APP Quantifications

Samples were collected using lancet finger-pricks (Microtainer Contact-Activated lancet, BD, Franklin Lakes, NJ, USA) dried on Whatman 903 Protein Saver DBS cards (Merck Sigma-Aldrich, Darmstadt, Germany). The cards were dried at 4 °C in the presence of a desiccant and processed daily.

We used the AbsoluteIDQ^®^ p180 Kit (Biocrates, Innsbruck, Austria) to quantify the concentrations of 21 amino acids (alanine, arginine, asparagine, aspartate, citrulline, glutamine, glutamate, glycine, histidine, isoleucine, leucine, lysine, methionine, ornithine, phenylalanine, proline, serine, threonine, tryptophan, tyrosine, and valine).

We used SISCAPA-MRM mass spectrometry to analyze selected acute-phase proteins as described [25] (Alpha-1-acid glycoprotein—A1AG11; Cystatin C—CST3; C-reactive protein—CRP; Hemoglobin beta chain—HBAA; Haptoglobin—HPTT; Insulin-like growth factor 1—IGF1; Lipopolysaccharide binding protein—LBP; Mannose-binding lectin—MBL2; Myeloperoxidase—MPO and Serum amyloid A1—SAA11).

### 2.3. Bioinformatic Analysis

Raw data were analyzed using the SciPy Python library (ver. 1.12.0) in Python 3.11.1. We used non-parametric Spearman’s rank-order correlation (rs, scipy.stats.spearmanr method) as in previous studies [23]. The off-diagonal entries represent direct positive or negative correlations of pairwise proteins (the correlation matrix is symmetric; diagonal entries represent self-correlations). Appendix A includes all data, but the discussion focuses only on protein pairs with ρ > 0.5 and significance of *p* < 10^–3^. Correlation matrices were visualized using Matlab R2023b (Mathworks, Natick, MA, USA) and the PyPlot Python library (ver. 3.1.2). To emphasize correlations with statistical significance, only correlations with *p* < 0.01 or *p* < 10^–3^ were used in the resulting image, as indicated. The color of each cell was then determined by linearly interpolating the correlation (blue for positive correlation, red for negative correlation).

Violin plots were generated using Matlab ver. 2023b (Mathworks, Natick, MA, USA) and grpandplot ver. 1.0.0 [44].

Sports-specific changes in AA and APP from pre- to post-exercise across all sports—cycling, karate, and modern pentathlon—were visualized as a network using NAViGaTOR ver. 3.0.19 [45]. The final figure with legends was prepared from an exported SVG file in Adobe Illustrator ver. 28.4.

## 3. Results

### 3.1. Acute-Phase Proteins

We measured and analyzed 10 proteins to examine the exercise-induced inflammatory acute-phase response and other markers among elite female athletes, highlighting the differences from their male counterparts. A comprehensive analysis of blood concentrations considering all collection states unveiled significant sex dimorphism for most of the measured APPs (Figure 1a,b). It is essential to highlight that CRP, LBP, HP, SAA1, MBL2, and MPO exhibited a broader range of concentrations in women than in men (Figure 1a and Figure 2a). Overall, IGF1, HP, LBP, MPO, and CRP (in ascending order of dimorphism impact) were higher in women, with a female-to-male median ratio varying from 1.2 to 2.0 (*p* < 0.001) across all states and sports (Figure 1b). In contrast, ORM1 and MBL2 were significantly lower in women (1.2-fold and 2-fold, respectively; *p* ≤ 0.001) (Figure 1a,b). Three proteins—SAA1 (*p* = 0.1), HBA (*p* = 0.075), and CST3 (*p* = 0.219)—were not different between women and men (Figure 1a).

Proteins linked to general inflammation (CRP), gut permeability (LBP), neutrophil activity (MPO), and hemolysis (HP) exhibited similar patterns of intersexual variations across states (Figure 2a). While significantly higher in women in most analyzed states, sex-based differences in APP appear to be attenuated by exercise. CRP concentrations were significantly higher in women during fasting and pre-competition (3-fold and 2.1-fold, respectively), whereas post-exercise collection states were more similar between sexes (~1.8-fold higher in women post-exercise, with no significant difference in resting) (Figure 2b). Although not statistically significant, SAA1 displayed a trend toward higher concentrations in women from fasting to post-exercise (1.4–1.1-fold) and lower concentrations in resting (1.5-fold), mirroring CRP’s sex-based kinetic differences (Figure 2b).

Exercise also attenuated sex-based differences in LBP, MPO, HP, and IGF1. LBP and MPO were significantly higher across all states and sports in women (1.5-fold and 1.4-fold, respectively) (Figure 1a and Figure 1b). During post-exercise states, LBP’s sex-based difference progressively decreased, remaining statistically significant post-exercise (1.4-fold), but no longer in resting (Figure 2a). MPO was higher in women pre-exercise, but exercise acutely attenuated the sex-based difference, which was no longer significant post-exercise, returning to the pre-exercise women-to-men ratio during resting (Figure 2a). HP’s women-to-men median ratio was also elevated from fasting to post-exercise, with no significant sex-based difference in resting (Figure 2a,b). IGF1’s response appeared to follow a similar pattern but with a smaller magnitude of intersexual difference (varying from 1.3-fold higher in fasting to 1.1-fold higher post-exercise, with no significant difference in resting) (Figure 2a,b).

In contrast, MBL2 and ORM1 were predominantly lower in female than in male athletes. MBL2 concentrations were not statistically significantly lower in the women during fasting and pre-exercise. Exercise induced a sex-based difference during post-exercise collections, from 1.8-fold post-exercise to 2.6-fold lower in women during resting. ORM1 was significantly different pre-exercise, and exercise increased the dimorphism, albeit to a lesser extent than it did for MBL2 (Figure 2a,b).

To explore potential relationships between specific APPs, we performed Spearman correlation analyses of the proteins, considering all states and sports together. We chose to discuss correlations with *p* < 10^–3^ and ρ > 0.5. Both female and male athletes presented significant positive correlations between CRP and SAA1, CRP and LBP, and SAA1 and ORM1 (ρ = 0.5–0.6) (Figure 3). In addition, women also presented significant positive correlations between SAA1 and LBP, MBL2 and ORM1, LBP and CST3, and CST3 and HBA (ρ = 0.5–0.6). All correlations can be found in Appendix A.

We also explored correlations within specific sports due to their different physical and skill demands. We chose to investigate potential correlations in modern pentathlon, cycling, and karate, as they represent different types of exercise in terms of intensity, duration, movements, and biomechanical aspects. As hypothesized, the three analyzed sports presented different APP correlations for both women and men (Figure 3 and Figure 4). As expected, correlations within specific sports were stronger than correlations across all sports combined. In cycling, the pair CRP-SAA1 was the only one that showed a positive correlation for women and men (ρ > 0.9, ρ > 0.7, respectively). Meanwhile, the only discordant pair was MBL2-ORM1, showing a negative correlation for women and a positive one for men (ρ > −0.7, ρ = 0.8) (Figure 3). Women exhibited two additional pairs with negative correlations, also involving ORM1 (ORM1-CRP ρ = −0.8 and ORM1-HBA ρ = −0.9). In karate, only LBP-SAA1 was correlated for both female and male athletes (ρ > 0.7, ρ > 0.8, respectively) (Figure 4). CRP-SAA1 was correlated across all analyzed sports for women but not for male athletes (Figure 3).

To emphasize changes in correlation among female athletes across states, we computed Spearman correlations within the four different states for modern pentathlon (Figure 4). Exercise produced an increase in significant correlations among APPs, with the most notable being SAA1-ORM1 (ρ = 0.8), SAA1-LBP (ρ = 0.8), and LBP-CRP (ρ = 0.7). In MP, we found two pairs with significant negative correlation involving IGF1. IGF1-MBL2 were negatively correlated in post-exercise and resting (ρ = −0.5, ρ = −0.7), while IGF1-CRP were negatively correlated in pre-exercise, post-exercise, and resting (ρ = −0.6, ρ = −0.8, ρ = −0.7). It is noteworthy that IGF1 in modern pentathlon (MP) tended to be negatively correlated with all other proteins, especially after the fasting period, even though it did not reach statistical significance based on our predetermined threshold for discussion (*p* ≤ 0.01) (Appendix A).

### 3.2. Amino Acids

We measured and analyzed 21 amino acids across the four examined states. Elite female athletes exhibited lower concentrations of most amino acids, except for glutamate and alanine, which had similar concentrations in both sexes across all sports and states (Figure 5a,b). The concentration of selected amino acids followed a similar pattern in both sexes but at different magnitudes.

We did not find significant sex-specific differences in blood alanine in any analyzed state (Figure 6a,b). Alanine increased roughly 1.4-fold after breakfast (from fasting to pre-exercise) for both groups and additionally significantly increased by 6–12% with exercise, remaining 30–40% above fasting concentrations in resting (Figure 6a). Feeding promoted a rise in glutamine concentration of 51% in women and 29% in men, which decreased in response to exercise. Glutamine was significantly lower in female athletes, although they appeared to show greater glutamine conservation during resting than men (0.83 vs. 0.77, compared with pre-exercise), also maintaining a 25% higher concentration for the amino acid compared to fasting (while men reached the fasting concentration) (Figure 6a). Blood glutamate was mostly unchanged by feeding or exercise in both groups of athletes and was not significantly different between the sexes (Figure 6a,b).

Although almost 30% lower in women (Figure 5b), BCAA showed a similar response to exercise in both sexes, while aromatic amino acids (AAAs; the sum of tyrosine, tryptophane, and phenylalanine) were consumed during exercise in men but not in women (Figure 6a,b). A decreased blood ratio between BCAA and AAA (Fischer ratio) may play a role in developing exercise-induced central fatigue. Although the Fischer ratio was significantly lower in women in every analyzed state, the acute exercise-induced decrease occurred only in male athletes and remained lower in resting. Another theory related to central fatigue development is the serotoninergic theory, which is related to the ratio between tryptophan and BCAA blood concentration. Female athletes did not present significant changes in this ratio, while male athletes exhibited an 11% decrease from pre- to post-exercise and 16% from pre-exercise to resting). Selected correlation pairs were maintained across the sex vs. state analysis (Figure 7). Among the BCAAs; valine concentration consistently exceeded that of isoleucine and leucine. Total BCAA exhibited strong correlations with each BCAA in every analysis conducted (ρ ≥ 0.9), mirroring the correlations observed among the individual BCAAs themselves (ρ ≥ 0.8). Interestingly, tyrosine, tryptophan, and phenylalanine were correlated with BCAAs for both women and men, and these correlations were maintained in cycling and karate for women but not for men. Arginine also correlated with BCAAs only for women (ρ = 0.7) but not for men. Blood amino acids involved in the urea cycle were highly affected. Arginine was greatly affected by breakfast in both groups. The amino acid concentration in the blood rose 1.5-fold in female and 1.4-fold in male athletes (Figure 6). Women conserved arginine more than men (85 vs. 53%). Ornithine showed a different pattern per group. In women, ornithine doubled in response to breakfast, returning to fasting concentrations after exercise (post-exercise and resting). For male athletes, ornithine rose by 40% after breakfast, with a slower decrease caused by exercise (a further 20% drop post-exercise, returning to fasting concentrations in resting). Like ornithine, exercise caused a drastic reduction in citrulline concentrations in women. Exercise almost doubled the sex-specific difference in ornithine concentration from 1.1- to 1.9-fold lower in women (also mainly driven by a decrease in women from 63.1 to 32.1 µmol/L). Also, it increased the difference in citrulline concentration from 1.3- to 1.4-fold lower in women. Aspartate and asparagine were slightly affected by either breakfast or exercise. Interestingly, histidine was one of the most affected amino acids showing sex dimorphism. Blood histidine concentration in elite female athletes showed a very distinctive pattern compared to their male counterparts. The amino acid rose 5.4-fold after breakfast, dropping to 40% compared to the pre-exercise condition, with an additional 16% drop during resting. Male athletes had a minor increase after breakfast followed by a delayed drop in histidine blood concentrations (22.4, 62.7, 58.2, and 28.3 µmol/L for fasting, pre-exercise, post-exercise, and resting, respectively). The substantial drop in the women-to-men ratio in the post-exercise phase was primarily driven by a substantial decrease in women’s histidine concentration, from 61.6 to 25.7 µmol/L (Figure 6). Serine decreased by 21% in elite female athletes, while it did not change in men.

Amino acid correlations were affected by sports modality. Interestingly, female athletes presented a considerably higher number of correlated pairs than men in HOLO analyses and the specific sports analyzed (cycling and karate) (Figure 7).

In MP, as observed in the previous APP analysis, exercise increased the number of correlated pairs. However, in the amino acid evaluation, we observed that the predominant factor driving the increase in correlated pairs was primarily dietary intake, occurring between the “fasting” and “pre-exercise” states (Figure 4).

## 4. Discussion

Here, we demonstrated that selected acute-phase proteins showed sex-specific differences in response to exercise stress in elite athletes, either in direction or magnitude, highlighting the importance of exercise as a model for understanding immunometabolism.

There are limited data on elite athletes’ biological parameters, partially due to the need for anonymity or to avoid revealing data that other competitors can use to improve performance [35]. Such data on women are even more scarce. Sharing data on elite athletes (both women and men) is critical for improving the understanding of metabolic and inflammatory responses, and thus advancing exercise science and pathophysiology. Here, we explored parameters in elite athletes during world-class training and competitions, comparing sex-dependent inflammatory and metabolic responses. Our data are reinforced by mass-spectrometry-based anti-doping control in all analyzed athletes, ensuring the presented results are not affected by drugs, which could alter metabolic responses. Moreover, we used DBS due to its advantages, such as being easy to collect and less invasive, requiring a small amount of blood, the cost of shipping/storage being significantly lower, analyte stability, and reduced risk of infection [46].

### 4.1. Acute-Phase Protein Response in Elite Female Athletes

CRP concentrations are widely evaluated in clinical settings to screen, diagnose, and monitor inflammatory conditions [47,48,49]. CRP analysis can provide insights into the impact of a specific exercise session on an individual and their recovery process [43,50]. It has been shown that blood CRP changes are affected by multiple, often conflicting factors, such as ethnicity, medications, age, and sex [51,52,53]. Women present higher CRP concentrations than men, even when adjusting for age, medication, and cardiovascular risk factors [54]. Other studies have reported higher CRP concentrations in women, with a stronger correlation between CRP concentration and central adiposity, with the difference being maintained across ethnic groups [55,56,57,58]. It is widely accepted that acute exercise can lead to a transient increase in serum CRP depending on factors such as exercise intensity and individual adaptation to the exercise [59,60]. Our results show that this sex-based difference is maintained across elite athletes, with women presenting higher concentrations of CRP (Figure 1a). However, the sex-based difference was attenuated during post-exercise periods. The CRP median women-to-men ratio progressively decreased from 3.1 in fasting to 1.1 in resting, being no longer different between sexes in the last analyzed state. Similarly, the SAA1 median women-to-men ratio decreased from 1.4 in fasting to 0.7 in resting. However, SAA1 was not significantly different between sexes in any state and only increased in male athletes from pre-exercise to resting. In fact, it has been reported that female soccer and netball athletes might present little acute-phase response to exercise under typical training challenges (not competition), with CRP likely being the most sensitive protein [26]. Also, our data suggest that women exhibit a broader range of CRP values (Figure 2a). The higher fasting values for women’s CRP and the different patterns in the post-exercise state seem valuable when tracking the impact of training and recovery. Tailoring training sessions and recovery protocols considering these sex-specific inflammatory responses might reduce the risk of overtraining or injury and improve recovery time (so important to elite athletes).

LBP is released in response to LPS entering the bloodstream, following the presence of bacteria or LPS translocation [61,62]. Intense exercise can increase epithelial wall permeability to LPS, supporting LBP as a marker for gut permeability changes during exercise [63,64]. These findings support the idea of exercise as a model for immunometabolism during metabolic stress. We have previously shown that LBP positively correlated with CRP under inflammatory conditions (~1700 DBS samples collected during infections, vaccinations, surgery, intense exercise, and Crohn’s disease) [30]. Unlike CRP, there are limited data on sex differences in LBP concentrations among the general population. Even though LBP has been increasingly used in clinical research, we could not find any relevant data on elite athletes. A recent systematic review showed no sex-based difference in indirect markers of gut damage or permeability following different types of exercise [65]. Unfortunately, LBP was not included in the meta-analysis [65]. Marriot et al. reported in an in vivo study a lower increase in LBP following intraperitoneal injections of LPS in female mice [66]. In that study, the authors collected samples 24 h after injection, which can be partially compared with our dataset’s “resting” state. Our data show that female athletes have significantly higher LBP concentrations across all states than men (Figure 1a), with an exercise-induced decrease in the sex-based difference, from 1.5 in pre-exercise (significantly different between sexes) to 1.2 in resting (not different). Moreover, our data showed that LBP correlates with CRP in women across all sports, with stronger correlations observed in modern pentathlon (a sport combining a long-duration exercise with different intensities and skills).

It is known that blood haptoglobin decreases in response to hemolytic conditions due to it forming complexes with free hemoglobin [67]. Exercise-induced hemolysis can be intensity-dependent due to mechanical or metabolic mechanisms, thus reducing haptoglobin concentration [68]. In our study, we did not find significant HP changes from pre- to post-exercise for both sexes; however, from pre-exercise to resting, HP significantly decreased in women and increased in male athletes (Figure 2a). Women also presented higher HP concentration across all states, with the median women-to-men ratio varying from 1.9 to 1.2 (fasting and pre-exercise/post-exercise, respectively). This finding suggests that elite female athletes may experience an opposite hemolysis response to exercise than men. It is critical to highlight that interpreting changes in the women-to-men ratio of HP is challenging since modifications can be due to hemolysis, inflammatory response, or their combination. HP concentrations can increase in reactive states, such as infection and trauma [69,70]. In fact, HP was not different between sexes in resting, similar to CRP and LBP response findings (Figure 2a). Myeloperoxidase (MPO) is an enzyme found in granulocytes, particularly in neutrophils and monocytes. Previously, we demonstrated that neutrophil count increase correlates with CRP in male amateur triathletes during a 200 km cycling race [71]. It has also been shown that different types of exercise and sports, as well as different intensities (e.g., 45% VO_2_ max/4 h, 60% VO2 max/3 h, and 75% VO_2_ max/2 h), increased MPO and neutrophil counts in male athletes [72,73]. However, it seems that MPO and neutrophil counts do not correlate in the post-exercise collection. However, the low number of participants and the absence of sex-specific analysis limit the study’s conclusions [73]. The increase in MPO in response to exercise has also been described in animal models [74]. Our data show that exercise acutely decreased the MPO women-to-men ratio from 1.4 in pre-exercise to 1.1 in post-exercise, returning to the pre-exercise ratio during resting. Also, MPO increased significantly in both sexes from pre- to post-exercise, but the elevation remained significant only among female athletes during resting. Although MPO did not correlate with any APPs measured across all sports, it did show a positive correlation with CRP (pre-exercise), MBL2 (post-exercise), and HP, LBP, and MBL2 (resting) in female modern pentathlon athletes (Figure 4). Conversely, MBL2 was approximately 2-fold lower in elite female athletes and significantly different in post-exercise collections (1.8-fold lower in women post-exercise and 2.6-fold lower in resting). The literature in the field is scarce, but a previous study indicated that healthy non-athlete women presented similar MBL2 values to men and did not identify changes in MBL2 induced by ~25 min of progressive-load exercise on a cycle ergometer [75].

We found that women exhibit a broader APP concentration range than men, which occurred without reported injuries or inflammatory events in the analyzed athletes. This broader APP response in women can influence sports and clinical translational studies. We reported evidence of sex-based differences in APP concentrations, with women appearing to be at a higher risk of suffering inflammation and, subsequently, injuries, as repeatedly described [60,76]. Sex-based differences must be considered in APP responses and N-of-1 trials for athletes are paramount to implementing tailored protocols (training, competition, resting, and rehabilitation). These trials are even more necessary considering that different sports may induce different immunometabolic responses, as shown in this study (Figure 8).

### 4.2. Amino Acid Metabolism in Elite Female Athletes

In addition to their structural role, amino acids can perform various metabolic and signaling tasks. Different models of metabolic stress can impact and be impacted by amino acid metabolism, such as sepsis, cancer, burn injury, hepatitis, diabetes, or obesity [77,78,79,80,81,82,83,84]. Exercise has been used as a suitable model for studying metabolic stress [13,35,85,86,87]. Amino acid metabolism has been mainly assessed by studies focusing on amino acid concentrations in the muscle or plasma [88]. However, geolocation and seasonal variations throughout the year have been shown to impact the amino acid pool during exercise [89]. Additionally, the microbiome can influence the general metabolite concentrations, including amino acids [64,90]. We analyzed sex-related ex post facto changes in elite athletes’ plasma amino acids from various sports during different metabolic states in a highly diverse country. This investigation may provide insight into the crucial role of amino acids during elite-level athletic performance.

Our data align with the current view, indicating that women’s amino acid concentrations are lower than men’s [91,92,93]. However, some studies have found controversial results regarding specific plasma amino acids in women of different ages and ethnic populations [94,95,96]. Alanine, glutamine, and glutamate significantly influence both anaplerosis and gluconeogenesis. Additionally, they play an essential role in ammonia metabolism, a key metabolite in exercise-induced central fatigue [14,16,97]. We did not detect sex-related differences in alanine blood concentrations in elite athletes. It has been shown in untrained individuals that alaninemia is altered by exercise in a duration-dependent manner, increasing in exercises lasting up to 80 min and decreasing in longer ones [98], a phenomenon that seems to mirror alanine concentration changes within the skeletal muscle [88,99]. We found a significantly higher increase in blood alanine in women (12% vs. 6%), as expected for non-prolonged exercise.

Our data did not show an exercise-induced change in glutamate concentrations in the blood, while elite female athletes exhibited a significantly lower decrease in glutamine. Glutamine can be depleted during extended exercise (like alanine), while it may increase after short bouts of exercise, especially with high output [100]. The decrease in glutamine blood concentration can be observed during prolonged exercise and other models of hypermetabolic stress, such as post-surgery and overtraining syndrome [101,102]. Glutamine plays an essential role as an ammonia transporter from muscle to the liver and is critical for exercise maintenance [97]. This finding may suggest a sex-based difference in the role of amino acids in the acute energy metabolism response during exercise. Due to glutamate’s pivotal role in energy metabolism within skeletal muscle, exercise can lead to blood glutamate depletion by increasing its uptake into muscle [103]. Here, we showed that blood glutamate concentration did not significantly change in female athletes, while it increased by 9% in men post-exercise. Indeed, we previously reported in a windsurf trial that glutamate was less affected by exercise than alanine and glutamine [104]. We confirmed that elite athletes presented minor changes in blood glutamate, alanine, and glutamine after dietary adjustments and training. More importantly, these amino acids were associated with a less fluctuating state, along with decreasing biomarkers of muscle injury, as previously shown by our group [104]. Elite female athletes presented significantly lower blood concentrations of BCAA in all collection states than their male counterparts. In our study, BCAA responses were highly correlated among the individual amino acids (ρ ≥ 0.9; *p* < 0.001) without differences between sexes in their response to exercise. Since BCAAs are ketogenic or glycogenic amino acids (or both), their role in metabolism depends on nutritional status and training. We did not observe an increase in BCAA response to exercise, probably because we analyzed changes induced by acute exercise bouts, as previously described [105,106]. Margolis et al. (2021) showed that BCAAs significantly increased before exercise following a glycogen-depleting protocol and a low-carbohydrate diet [107]. In addition, lower BCAA concentrations in women have been observed in the military during basic combat training, while ten weeks of training caused an increase in both BCAA and total amino acid concentrations in both sexes, with higher levels observed in women among the military population [108].

As previously described in animal models and the general population, we also found that exercise decreased the Fischer ratio for both female and male elite athletes, favoring the development of central fatigue [109,110]. However, our data show that elite female athletes had a smaller decrease in Fischer ratio in response to exercise, although they also presented a significantly lower ratio pre-exercise. Women’s Fischer ratios did not change acutely post-exercise. Still, they significantly decreased by 5% from pre-exercise to resting, while men’s Fischer ratio decreased by 9% already in post-exercise and then 10% from pre-exercise to resting. In addition, BCAA and AAA had a high correlation in female athletes for cycling and karate but not in men, suggesting sex-based differences in exercise metabolism during specific sports (Figure 7). To the best of our knowledge, this is the first study showing sex-based differences in the Fischer ratio during exercise. This finding can be of importance for understanding exercise metabolism, as well as the role of BCAA/AAA in liver failure and encephalopathies.

In the 1980s, Newsholme proposed a possible link between the increased tryptophan/BCAA ratio and central fatigue development, now commonly referred to as the “serotonin hypothesis”, since tryptophan is a serotonin precursor [18,111]. We did not detect exercise-induced changes in tryptophan blood concentration in female or male athletes; however, women presented a significantly higher tryptophan/BCAA ratio in pre-exercise collections. In addition, elite female athletes did not exhibit a significant exercise-induced increase in the tryptophan/BCAA ratio, while men showed a significant increase of 11% from pre- to post-exercise and 16% from pre-exercise to resting. While prolonged exercise can decrease blood BCAA concentrations due to potential utilization as an energy source within muscles or hepatocytes, prolonged exercise may increase blood free-tryptophan concentrations. While the influence of tryptophan on central fatigue remains controversial, clinical studies have been evaluating it. Maciejak et al. observed that intragastric administration of FFAs increased the seizure threshold and induced sedation, an effect abolished when tryptophan passage into the brain was blocked [112].

We highlight that sex comparisons regarding central and peripheral fatigue development have attracted the scientific community’s attention [31,113,114,115]. However, data retrieved from elite athletes have not been published. After evaluating clinical tests, Jo et al. proposed that women presented a trend of being less affected by exercise-induced central fatigue than men following sustained isometric ankle plantar flexion [116]. Together with our analysis regarding the Fischer ratio and tryptophan/BCAA ratio, we add that elite female athletes exhibit weaker exercise-induced amino acid responses, which have been evaluated as indirect central fatigue markers. However, our data showed that female athletes can be at a higher risk for developing central fatigue.

### 4.3. The Immune-Metabolic Response

The decrease in glutamine concentration and subsequent availability has been linked to the decline in immune function due to the energetic needs of both intestinal and white blood cells for glutamine as a fuel. This effect was seen in the four individually analyzed sports and was concurrent with increased MBL2 and MPO in cycling.

It is interesting to highlight that the metabolism of BCAA, either collectively or by sport, was different and was not associated with a particular immune response. Otherwise, the relationship between the urea cycle intermediates was observed across all sports and MPO [117]. Citrulline has been related to inflammatory response, including during rheumatoid arthritis [118]. The relationship between the NO precursors and MPO has been discussed, and the enzyme’s activity can be related to different causes of joint inflammation [119]. These events relate to our previous idea of differentiating HP increases due to hemolysis from those caused by inflammation [120]. The influence of amino acid metabolism and its integration with the immune response can be of great interest for understanding the response to hypermetabolic states, as we described previously [8].

## 5. Conclusions

Here, we demonstrate that the sex-based disparity in APP concentrations in each state appears more critical than the sex-based difference in the APP response.

Our data reveal significant sex-specific and sex-agnostic correlations between proteins and amino acids across specific sports. This emphasizes the need for investigations focusing on sex-based differences in metabolism across different sports or even grouping sports by similarities (e.g., intensity, duration, power output). Previous evaluations on sex differences during fatigue have relied on controlled exercises rather than field-of-play situations, which can limit data translation to elite athletes. Moreover, we suggest the need for more molecular investigations to understand sex differences in central fatigue development, utilizing other markers.

Taking the results together, we propose the importance of exercise as a model for understanding immunometabolism in physiological and pathological conditions.

## 6. Limitations

This study has several inherent limitations. First, as an ex post facto sportomics analysis conducted in real-world training and competition settings, we had no control over various conditions, which introduced variability in factors such as diet, sports activity, intensity, and athlete age. Another limitation is the broad scope of the study, which covered multiple sports and a wide range of analytes. While this provides a comprehensive overview, it lacks the specificity that could be achieved in controlled experiments. These confounding factors limit the external validity of the findings and hinder the ability to infer causality. Therefore, the study’s results should not be used to discuss mechanisms or draw definitive conclusions, but rather to observe real differences in a non-controlled environment, raise new hypotheses for future investigation, or reinforce previously published data.

## Figures and Tables

**Figure 1 nutrients-16-03538-f001:**
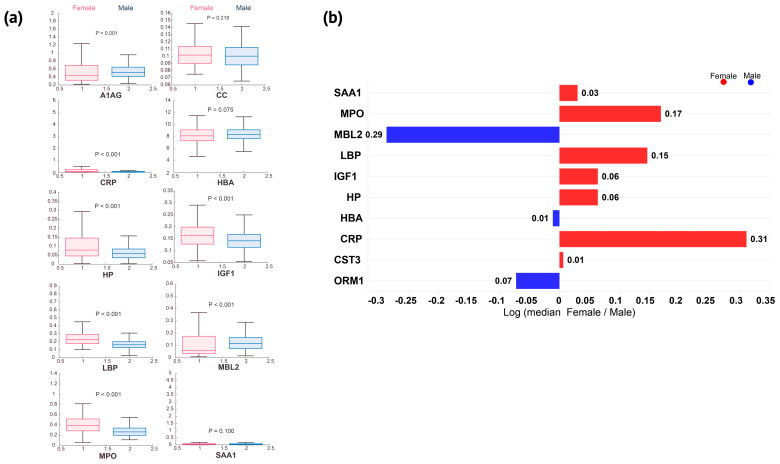
Female athletes presented higher concentrations of inflammatory proteins than male athletes. (**a**) Protein concentration distribution and comparison between sexes across all states and sports unveiled significant sex dimorphism. (**b**) Ratio of the median values for each measured protein between women and men, considering all states together.

**Figure 2 nutrients-16-03538-f002:**
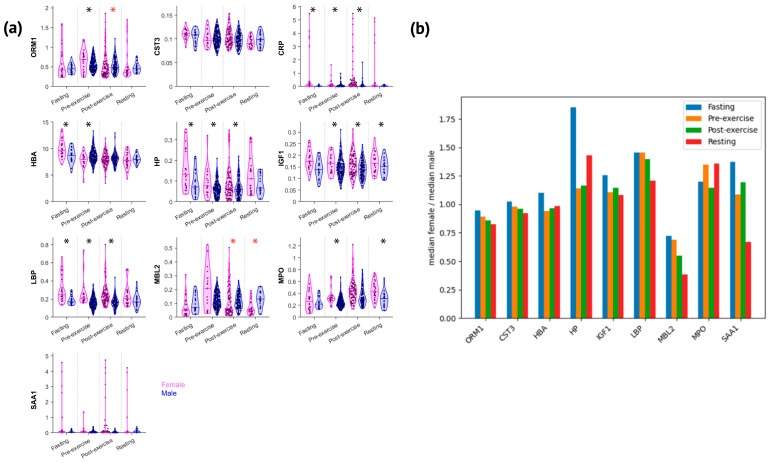
Female athletes presented a broader range of acute-phase protein concentrations than men in different states but seemed to exhibit a lower acute-phase response following exercise. (**a**) Protein changes and distribution across states adjusted for sex. (**b**) Ratio of the median values for each measured protein between women and men across all analyzed states. * = significantly higher in women. * = significantly lower in women.

**Figure 3 nutrients-16-03538-f003:**
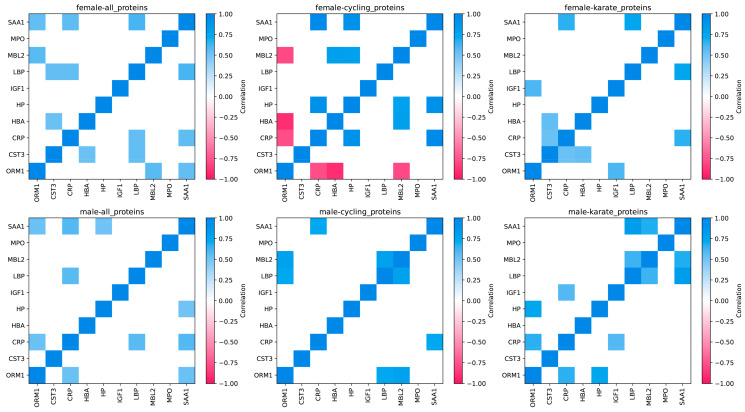
Spearman pairwise correlation matrices for proteins, considering all states together across different sports. Only correlations with *p* < 0.001 and ρ > 0.5 are highlighted.

**Figure 4 nutrients-16-03538-f004:**
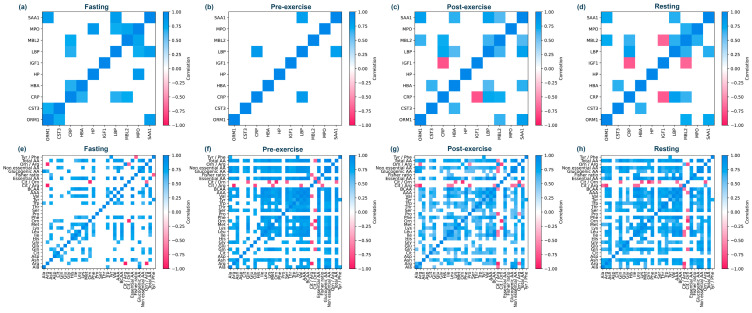
Spearman pairwise correlation matrices for proteins and amino acids, considering samples collected from female modern pentathlon athletes. Panels (**a**–**d**) show the measured protein correlations in different states; panels (**e**–**h**) show the amino acids correlations at different exercise moments. Only correlations with *p* < 0.001 and ρ > 0.5 are highlighted.

**Figure 5 nutrients-16-03538-f005:**
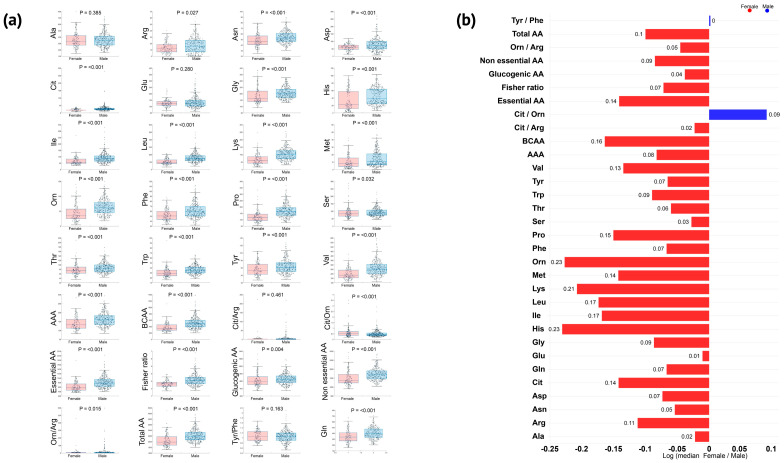
Women present lower amino acid blood concentrations than male athletes. (**a**) Amino acid distribution and comparison between sexes across all states and sports unveiled significant sex dimorphism. (**b**) Ratio of the median values for each measured amino acid between women and men considering all states together.

**Figure 6 nutrients-16-03538-f006:**
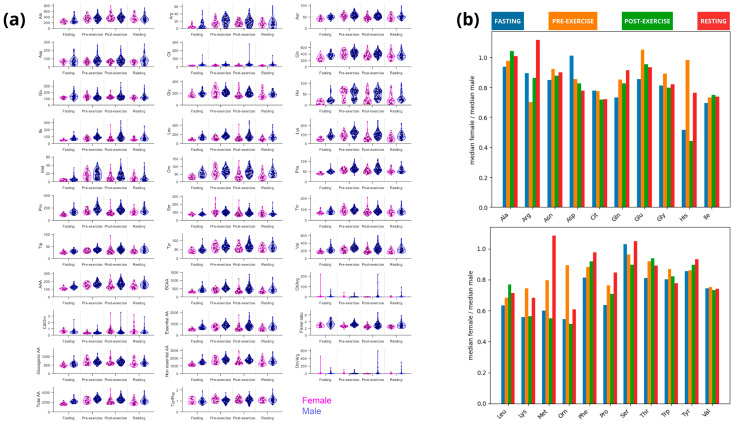
Selected amino acids involved in energy metabolism, protein synthesis, recovery, and central fatigue exhibit sex-based differences in response to exercise. (**a**) Amino acid kinetics and distribution across the states for each analyzed sex. (**b**) Ratio of the median values for each measured amino acid between women and men across all analyzed states.

**Figure 7 nutrients-16-03538-f007:**
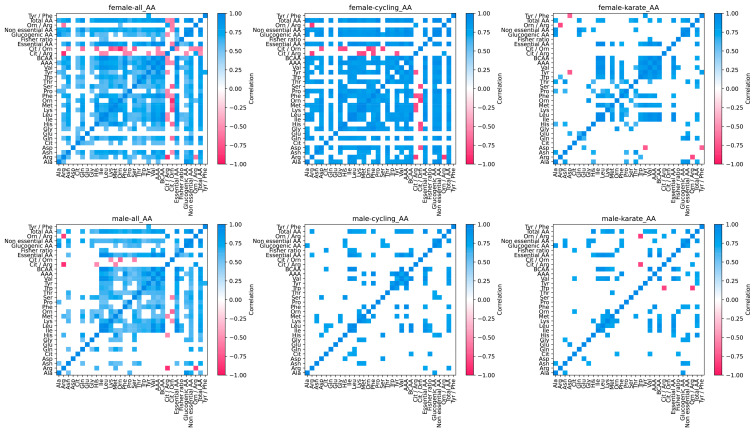
Spearman pairwise correlation matrices for amino acids, considering all states together across different sports. Only correlations with *p* < 0.001 and ρ > 0.5 are highlighted.

**Figure 8 nutrients-16-03538-f008:**
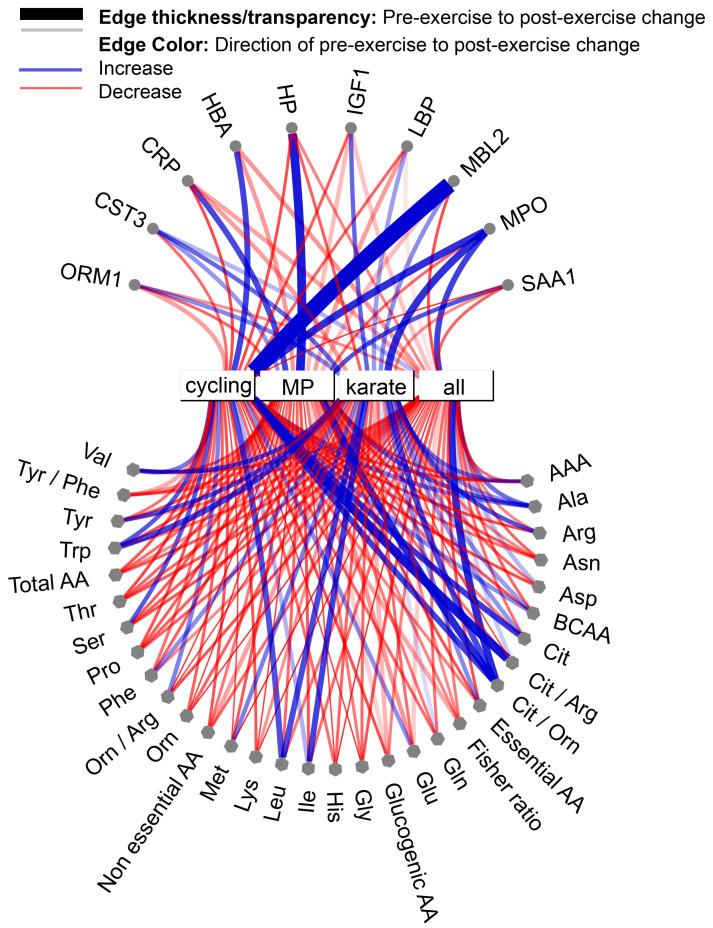
Chord diagram showing exercise-induced trend changes from pre-exercise to post-exercise for each analyzed analyte in female samples across cycling, karate, modern pentathlon, and all sports combined. Node shape signifies AAs and APPs, while edge thickness/transparency represents the change, and color shows the direction of the change pre- to post-exercise, as per legend. Visualized in NAViGaTOR.

## Data Availability

The data supporting this study’s findings are available from the corresponding author upon reasonable request.

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
