# Peer review of "Sportomics Analyses of the Exercise-Induced Impact on Amino Acid Metabolism and Acute-Phase Protein Kinetics in Female Olympic Athletes"

_nutrients, 2024, doi:10.3390/nu16203538_

Round 1

Reviewer 1 Report

Comments and Suggestions for Authors

Congratulations for the paper.  Overall, the text of the paper is very extensive. Mainly in results and discussion. In fact, there are too many references. The authors should make an effort to synthesize the most important information.

My major comments are:

·       The last author is missing

·       Authors should use "women" and "men" throughout the text.

·       Line 32: The sentence must be written differently. For example, "We compared 447 women and 990 men elite athletes"

·       Line 35: should say "women elite athletes"

·       The objective of the study at the end of the introduction section should be better stated. How many objectives does the study have? Two main objectives, one primary objective and one secondary objective?

·       Authors should review the verb tense throughout the text. There are sentences that should be in the past tense. For example, line 147-148

·       Line 182-184: These lines should go in the method. If the information is repeated, they should be deleted.

·       The figures are very small, you can't see the information

·       In the results section, authors should explain the data they obtained. They should not repeat information that is in the method section.

·       The discussion section should begin with the main finding of the study.

·       verb tense? verb in the past tense?

·       The discussion is too long. The authors should present their data and compare it with the existing literature. 

·       The conclusion section is missing. The authors should write a paragraph of 2-3 sentences. They should write a clear message

·       The style of reference is not correct.

·       The number of references is excessive.

Author Response

Comment 1: Congratulations for the paper. Overall, the text of the paper is very extensive. Mainly in results and discussion. In fact, there are too many references. The authors should make an effort to synthesize the most important information.

Response: We thank reviewer 1 for the compliment. Also, we agreed with reviewers 1 and 2 and shortened the paper.

Comment: My major comments are:

The last author is missing

Response: We wondered if reviewer 1 received the paper after the editorial team's first review round. The names in the submitted PDF were correct, and there was a conclusion.

Comment:Authors should use "women" and "men" throughout the text.

Line 32: The sentence must be written differently. For example, "We compared 447 women and 990 men elite athletes".

Line 35: should say "women elite athletes".

Response: Thank you for the suggestion, we have corrected the entire manuscript accordingly.

Comment: The objective of the study at the end of the introduction section should be better stated. How many objectives does the study have? Two main objectives, one primary objective and one secondary objective?

Response: We have revised this paragraph to improve clarity:

“Our main objective was to investigate AA and APP's responses in Olympic athletes from 16 Olympic sports across different states (fasting, pre-exercise, post-exercise, and resting) during training sessions and competitions. By performing sex-based stratification, we aimed to identify and characterize potential sex-specific differences observed and present all quantitative data. Due to the similarity of metabolic and inflammatory pathways in exercise and certain pathological conditions, our results may provide a better understanding of these responses in different scenarios.”

Comment: Authors should review the verb tense throughout the text. There are sentences that should be in the past tense. For example, line 147-148

Response: We carefully double-checked the verb tense.

Comment: Line 182-184: These lines should go in the method. If the information is repeated, they should be deleted.

Response: We thank the reviewer for your careful reading of the manuscript. We altered it according to your suggestion.

Comment: The figures are very small, you can't see the information

Response: Unfortunately, the Nutrients editorial system requires that figures be embedded within the manuscript, which limits our control over their size. However, all figures are high resolution and can be provided individually upon request.

Comment: In the results section, authors should explain the data they obtained. They should not repeat information that is in the method section.

Response: We thank the reviewer for careful reading of the manuscript. We altered it according to your suggestion.

Comment: The discussion section should begin with the main finding of the study.

verb tense? verb in the past tense?

Response: We included the main finding in the beginning of the discussion section, according to your suggestion. We carefully double-checked the verb tense.

Comment: The discussion is too long. The authors should present their data and compare it with the existing literature. 

Response: Also, we agreed with reviewers 1 and 2 and shortened the paper.

Comment: The conclusion section is missing. The authors should write a paragraph of 2-3 sentences. They should write a clear message.

Response: We are not sure if reviewer 1 received the paper after the editorial team's first review round. There was a conclusion. We are resending it. In addition, we made significant changes in the manuscript in order to improve its clarity.

We added a limitations section, describing a possible bias on different variables such as diet, health status, sports activities, sports intensities, and age. But briefly, we highlight that we were dealing with athletes in the most important sports events in the world, which led us to different idiosyncrasies. For example, in a specific case we could not collect samples from a pair of athletes after the match because they were celebrating with parents, friends, and the press after winning the gold medal.

Comment: The style of reference is not correct.

Response: We thank the reviewer for your careful reading of the manuscript. We altered the reference format.

The number of references is excessive.

Response: We thank the reviewer for your careful reading of the manuscript. We have reduced the number of references.

Reviewer 2 Report

Comments and Suggestions for Authors

thanks a lot for the opportunity to score this work that has the  merit to investigate integender differences in some metabolites in elite athletes but as structured in its analysis is at high risk of bias. 

The work is descriptive of the intergender differences in acute phase proteins, and amino acids in elite athletes masured in a large sample of subjects. The main limitation of the work is the lack of a proper control of the population of athletes regarding the ovrerall condition including diet and health status. This flaw may severely impact on results. Indeed, there is no normalization for the sport activities and sport intensity (this can impact at least on acute phase proteins). Further, the sport activities considered in males and females are different. There is no stratification for age. There is no control of the time of blood collection in fasting condition and after exercise (this may significantly impact on acute phase proteins). 

From this point of view discussion is extremely speculative.

Overall this work should be rewritten as mere description of the differences in two uncontrolled population whithout flight of fancy in the discussion section and with a large limitation section unless a proper stratification of data has been done.

Author Response

Comment: 

Thanks a lot for the opportunity to score this work that has the merit to investigate intergender differences in some metabolites in elite athletes but as structured in its analysis is at high risk of bias. The work is descriptive of the intergender differences in acute phase proteins and amino acids in elite athletes measured in a large sample of subjects. The main limitation of the work is the lack of proper control of the population of athletes regarding the overall condition, including diet and health status. This flaw may severely impact on results. Indeed, there is no normalization for the sport activities and sport intensity (this can impact at least on acute phase proteins). Further, the sport activities considered in males and females are different. There is no stratification for age. There is no control of the time of blood collection in fasting condition and after exercise (this may significantly impact on acute phase proteins). From this point of view discussion is extremely speculative. Overall this work should be rewritten as mere description of the differences in two uncontrolled population whithout flight of fancy in the discussion section and with a large limitation section unless a proper stratification of data has been done.

Response:

We agree with Reviewer 2 on many aspects related to rationale and are genuinely grateful for the constructive criticism and comments.

Our aim was never to discuss mechanisms or causality based on the results presented here, and we have thoroughly reviewed the text to remove any phrasing that might have unintentionally suggested otherwise.

It is important to emphasize that this study was not designed as a controlled experiment but rather as an ex-post-facto sportomics analysis in the field of play (training or competition) across various sports, with the analysis stratified by sex. Therefore, the goal is to describe these differences and what they may imply rather than discuss why they occurred, whether due to sport, diet, or other factors.

With all due respect to the comment about the study, we would like to emphasize that one goal was to guide the athletes during the games, but we are not pursuing a final "conclusion" in this manuscript. As scientists, we do not believe in absolute or final conclusions and are trying to provide insights that shape possibilities for future studies. Since our research unveils results from top-level athletes, its results open novel findings in extreme training and competition moments. Our study has broad implications because that is what we are pursuing. In addition, studies with elite athletes can be useful for the understanding of physiological or pathological conditions, and these inflammatory responses are, precisely, broad. This may have many positive implications in disease management, as described by us previously (França TCL et al., doi: 10.3389/fnut.2023.1169188).

Concerning the lengthy discussion, we briefly introduce the biological significance of the measured molecules in sports and, when available, reference studies that have evaluated sex differences. Even beyond the scope of this study to explain how these differences arose or to assert their definitive meaning, we focused on describing the findings and their potential biological relevance within sports science.

We appreciate the reviewer’s feedback and have clarified this approach by removing speculative causal and mechanistic interpretations (even though in the original text, we used qualifiers such as “may” or “potentially”) while retaining descriptions and potential relevance (also moderated or omitted where necessary) to draw attention to the possible implications that warrant further investigation. In summary, we significantly reduced the length of discussion and references.

Studies with elite athletes as subjects generally do not have large numbers, especially in sports such as windsurfers, ultra-endurance cyclists, and others, where we see papers with N of 1, for example: Bessa et al., 2008 (10.1136/bjsm.2007.043786), Resende et al., 2011 ( 10.1089/omi.2011.0010), Merrit et al., 2019 (10.3389/fphys.2019.01410), Schork 2022 (10.1162/99608f92.f1eef6f4). The uniqueness of our study stems from evaluating almost a hundred elite athletes in the most significant sporting events in the world. We are not trying to elucidate mechanisms but to illuminate possible correlations in the studied proteins. We want to clarify to the reviewer that our study was an ex post facto study. We did not interfere with the conditions. We did not perform a laboratory-controlled study, as when studying petroleum in petroleomics, we tried to understand general responses to facts that happened with minimal interference (as in astrophysics studies, we did not and could not control all the variables). Yes, we performed a study with many variables, and even with many variables, we were able to extract possible pathways and interactions that future studies can further elucidate.

Most papers in exercise science have studied middle-class athletes or active people (non-professional athletes). Little data are available on world-class competing athletes due to the necessity of keeping physical data secret to eliminate their use by competitors. Thus, the scientific community will enormously benefit from access to the metabolic information of top-level athletes. We write about contexts where the protein correlations we identified have been previously acknowledged or described. Dealing with protein correlations can be very broad, and no one can (or should) speculate about causality, so we carefully deleted double-meaning sentences.

We added a limitations section, including the possible bias on different variables such as diet, health status, sports activities, sports intensities, and age. But briefly, we highlight that we were dealing with athletes in the most important sports events in the world, which led us to different idiosyncrasies. For example, in one specific case we could not collect samples from a pair of athletes after the match because they were celebrating with parents, friends, and the press after winning the gold medal.

New Limitation Section:

“This study has several inherent limitations. First, as an ex-post-facto sportomics analysis conducted in real-world training and competition settings, we had no control over various conditions, which introduced variability in factors such as diet, sport activity, intensity, and athlete age. Another limitation is the broad scope of the study, which covered multiple sports and a wide range of analytes. While this provides a comprehensive overview, it lacks the specificity that might be achieved in controlled experiments. These confounding factors limit the external validity of the findings and hinder the ability to infer causality. Therefore, the study's results should not be used to discuss mechanisms or draw definitive conclusions, but rather to observe real differences in a non-controlled environment, raise new hypotheses for future investigation, or reinforce previously published data."

Round 2

Reviewer 1 Report

Comments and Suggestions for Authors

My minor comments are:

·       The authors name style is not correct.

·       The reference style is not correct

·       The figures are very small, you can't see the information

Author Response

We appreciate your meticulous and careful review of our article.

Comment: The authors name style is not correct.
Response: Regarding the style of the authors' names, could you kindly point out where the error lies? We have checked all the names, and they are in line with how we have always published, including recently in Nutrients in the article titled "Caffeine Boosts Weight-Lifting Performance in Rats: A Pilot Study", Nutrients 202416(13), 2022; https://doi.org/10.3390/nu16132022.

Comment: The reference style is not correct.
Response: As for the reference style, we have revised and adjusted it according to the journal's request (below). If any systematic or occasional errors remain, we kindly ask you to point them out to us. Thank you.

"References should be described as follows, depending on the type of work:

  • Journal Articles:
    1. Author 1, A.B.; Author 2, C.D. Title of the article. Abbreviated Journal Name Year, Volume, page range."

    As encouraged by the journal we also added the DOI, when available.

    "DOI numbers (Digital Object Identifier) are not mandatory but highly encouraged."

Comment: The figures are very small, you can't see the information.
Response: Regarding the size of the figures, we have removed unnecessary white space around them and are uploading the figures individually, separate from the article, to facilitate visualization.

Reviewer 2 Report

Comments and Suggestions for Authors

I appreciated the changes. I think the ms is significantly improved.

Author Response

Comment: I appreciated the changes. I think the ms is significantly improved.
Response: We thank the reviewer for their active participation and valuable insights in helping to improve our paper.